# Genetic Susceptibility to Human Norovirus Infection: An Update

**DOI:** 10.3390/v11030226

**Published:** 2019-03-06

**Authors:** Johan Nordgren, Lennart Svensson

**Affiliations:** 1Division of Molecular Virology, Department of Clinical and Experimental Medicine, Linköping University, 58185 Linköping, Sweden; johan.nordgren@liu.se; 2Division of Infectious Diseases, Department of Medicine, Karolinska Institute, 17177 Stockholm, Sweden

**Keywords:** norovirus, histo-blood group antigens, susceptibility, secretor status

## Abstract

Noroviruses are the most common etiological agent of acute gastroenteritis worldwide. Despite their high infectivity, a subpopulation of individuals is resistant to infection and disease. This susceptibility is norovirus genotype-dependent and is largely mediated by the presence or absence of human histo-blood group antigens (HBGAs) on gut epithelial surfaces. The synthesis of these HBGAs is mediated by fucosyl- and glycosyltransferases under the genetic control of the *FUT2* (secretor), *FUT3* (Lewis) and *ABO(H)* genes. The so-called non-secretors, having an inactivated FUT2 enzyme, do not express blood group antigens and are resistant to several norovirus genotypes, including the predominant GII.4. Significant genotypic and phenotypic diversity of HBGA expression exists between different human populations. Here, we review previous in vivo studies on genetic susceptibility to norovirus infection. These are discussed in relation to population susceptibility, vaccines, norovirus epidemiology and the impact on public health.

## 1. Introduction

Infectious diseases have influenced the evolution of the human genome, in part by selecting for host alleles that modify infection and pathogenesis. The concept that some individuals are more or less susceptible to infections is not new but, until relatively recently, it was assumed that the clinical outcome of an infection was mainly dependent on the virulence factors of the microorganism. The host genetic factors involved in innate or adaptive immunity or the expression of pathogen-specific receptors have received relatively little attention. A remarkable example of host genetic susceptibility is strong resistance to the most common noroviruses among individuals with a nonsense mutation in chromosome 19, a trait that exhibits a Mendelian pattern of inheritance.

Noroviruses, which cause the so-called winter vomiting disease, are the most common etiological agent of acute gastroenteritis, and are responsible for approximately 20% of all cases of acute gastroenteritis worldwide [1]. These viruses cause an estimated 200,000 child deaths each year, and are now the most common etiological agent of pediatric diarrhea in rotavirus-vaccinated populations [2,3,4,5,6]. Norovirus is highly infectious, with as few as 10 infectious particles required to cause disease [7]. The high numbers of virions in feces and vomitus, combined with high environmental stability and a low infectious dose, make norovirus highly transmittable and contagious, with many challenges for infection control. Norovirus outbreaks are frequently reported in hospitals, communal settings, nursing homes, and day care centers, and contribute significantly to foodborne outbreaks [8]. Long hampered by the lack of robust in vitro cell culture systems, efforts to develop effective norovirus vaccines are now progressing with the use of virus-like particles (VLPs), which is regarded as a promising approach to the control of norovirus burden [9,10].

In spite of the high infectivity and rapid transmission of norovirus, a subpopulation of individuals does not develop symptomatic disease or become infected after exposure. Studies have shown this phenomenon to be largely dependent on human histo-blood group antigens (HBGAs), mainly the types controlled by the *FUT2* (Secretor), *FUT3* (Lewis), and *ABO* genes. In vivo and in vitro studies have shown that the resistance is norovirus genotype- and even strain-dependent [11,12].

Here, we review previous in vivo studies on norovirus infections in relation to host genetic susceptibility. These are discussed in the light of population susceptibility, norovirus epidemiology and the impact on public health.

## 2. Norovirus—A Family with Many Members

Norovirus is a genus in the family *Caliciviridae*. The norovirus genome is organized into three open reading frames (ORFs), with ORF2 (approximately 1.8 kb in length) encoding the 57 kDa major capsid protein, VP1. The major capsid protein is further divided into three domains: the N-terminal domain, the intermediate shell domain, and the protruding P domain [13]. The P domain is subdivided into two major sub-domains, P1 and P2. Most of the cellular receptor interactions and immune recognition epitopes are located in the P2 sub-domain, extending above the viral surface, which is also the most variable part of the norovirus genome [14,15]. Several in vitro binding studies have demonstrated that the P2 domain is essential in determining host range [16,17].

The norovirus genus, which is highly diverse, is divided into seven different genogroups, of which GI, GII, and GIV cause disease in humans [18,19,20]. These genogroups exhibit high diversity, with up to 60% difference in the amino acid sequences of the major capsid gene [18]. The genogroups can be further divided into genotypes based on sequence diversity of the capsid gene, with at least nine for GI and 25 for GII having been described [19,21]. The most frequently occurring genotype, GII.4, is often further divided into specific variants, with new emerging variants often accounting for the recurrent seasonal pandemics [19,22,23,24]. Seven different GII.4 variants have been associated with global epidemics since the 1990s, occurring in 1996 (Grimsby), 2002 (Farmington Hills), 2004 (Hunter), 2007–2008 (2006a/Yerseke, 2006b/Den Haag), 2009–2012 (New Orleans), and 2012 and beyond (Sydney). These events indicate that new variants of GII.4 have appeared on average every 2 to 3 years [9].

## 3. A Brief History of Host Genetics and Norovirus Susceptibility

The first norovirus strain to be isolated was detected in a school outbreak in the town of Norwalk, USA. In the 1970s, this strain was used in a challenge study on 12 volunteers [25]. Interestingly, 50% of the individuals did not develop symptoms of norovirus gastroenteritis and, when challenged again 27 and 42 months later, these individuals remained asymptomatic while the other 50% of participants developed symptoms at each challenge. After 4–8 weeks, the symptomatic individuals were challenged again and only one developed symptoms. Most symptomatic individuals had increased serum antibody titers after each challenge. Taken together, these findings indicated that there was no long-lasting immunity to norovirus. A short-term immunity was noted, and factors other than serum antibodies appeared important for immunity. However, the virus challenge dose was many folds higher than the amount of virus that can cause illness. Thus, the perceived lack of long-term immunity may not be relevant for natural settings and infectious doses [26]. A subsequent study in 1982 indicated familial clustering of resistance after swimming exposure, suggesting involvement of host genetic factors [27]. Lindesmith et al. [28] provided direct evidence that host genetic factors contribute to susceptibility to norovirus by showing that secretor-negative individuals were resistant to Norwalk virus in a challenge study. In support of this observation, it was demonstrated that a VLP of the Norwalk strain bound to surface epithelial cells of the gastroduodenal junction as well as to saliva, but only in the so-called secretor-positive individuals, who express H antigens in saliva and mucosa [29]. Subsequent studies further confirmed the secretor status mediated by the *FUT2* gene as an important mediator of susceptibility [30]. Studies also showed that blood groups affect susceptibility, with blood group B, for example, offering partial protection against certain norovirus strains [29]. Thus, these studies identified human HBGAs, including the secretor, Lewis and ABO families, as important host factors mediating susceptibility and resistance to norovirus.

## 4. Genetic Control of HBGA Synthesis

The ABO(H) and Lewis carbohydrate structures are synthesized by the sequential addition of monosaccharides to precursor oligosaccharides that constitute the peripheral region of glycolipids as well as the O- and N-linked glycans of glycoproteins. The expression of α1,2-linked fucose residues on the surface epithelial cells of the gut and in body fluids (hence the term secretor) is dependent on the activity of the FUT2 enzyme, which catalyzes the synthesis of the H antigen (Figure 1). FUT2 is one of two active human α1,2-fucosyltransferases, the other being FUT1, encoded by the *FUT1* gene. The *FUT2* gene is expressed predominately in epithelial (mucosal) tissues, whereas the *FUT1* gene is typically expressed in erythropoietic progenitor cells. Human HBGAs controlled by FUT2 are, therefore, mainly present on mucosal epithelia of the respiratory, genitourinary and digestive tracts and as free oligosaccharides in body fluids such as blood, saliva, and milk. The *FUT3* gene encodes an α1,4-fucosyltransferase that converts the H type 1 antigen to Lewis b through the addition of fucose. Similarly, either an acetylgalactosamine or a galactose can be added by the A or B enzymes, encoded by the *A* and *B* alleles of the *ABO* gene, respectively (Figure 1). Individuals with a non-functional FUT2 are termed non-secretors, due to the absence of ABO(H) groups in saliva and mucosa. Thus, a non-secretor having blood group A (on the erythrocytes) will not express the A antigen in mucosal tissues. These individuals express Lewis a if they have a functional FUT3 enzyme which catalyzes the addition of an α1,4-linked fucose residue to the H type 1 precursor. Homozygotic inactive *FUT3* gene carriers thus lack Lewis a and b structures and are termed Lewis-negative. 

In the coding region of the *FUT2* gene, several missense and nonsense mutations that affect secretor status have been identified, many of which are population-specific (Table 1). The two most common and widely investigated in norovirus susceptibility studies are Se^428^ (resulting in a non-secretor phenotype) and Se^385^ (resulting in a weak-secretor phenotype) (Table 1).

## 5. Norovirus Challenge Studies

Challenge studies conducted with volunteers have been performed with three different genotypes of norovirus, the prototype Norwalk strain (GI.1 genotype), the snow mountain virus (SMV) strain (GII.2 genotype) and the globally predominant GII.4 genotype (strain Cin-1, variant Farmington Hills) [28,33,34,35,36,37]. All challenge studies using the Norwalk strain showed that non-secretors were protected against both symptomatic and asymptomatic infections, with individuals of blood group B also exhibiting a partial protection (Table 2). In contrast, a challenge study using the SMV strain (GII.2 genotype) showed no association of secretor status or ABH group antigens with susceptibility. This lack of association suggests the existence of an alternative primary ligand by which SMV infects the host; this has also been indicated in in vitro binding studies of several other norovirus genotypes [38]. A limitation of the Norwalk and SMV strains for use in challenge studies is that these genotypes are rarely found in natural infections; thus, results from these studies do not translate well in terms of clinical relevance. However, a more recent challenge study by Frenck et al., using a strain of the predominant GII.4 genotype, showed that the non-secretor phenotype was significantly associated with protection, although one individual with the non-secretor phenotype became asymptomatically infected [33]. However, this challenge study was only performed with one variant of the GII.4 genotype, and the results cannot be directly translated to other GII.4 variants, including the recently emerging variants.

Thus, the challenge studies have demonstrated variation in susceptibility patterns to different norovirus genotypes, with the prototype GI.1 and globally dominating GII.4 genotypes predominately infecting secretors. In contrast, the GII.2 SMV strain infects secretors and non-secretors alike, indicating that infection is facilitated by other ligands.

## 6. Norovirus Outbreak Studies

Much information regarding genetic susceptibility to norovirus infection has been obtained from outbreak studies involving single norovirus genotypes. Being the predominant outbreak genotype, GII.4 has been by far the most studied genotype (Table 3). Studies of outbreaks in various settings such as hospitals, nursing homes and military units have been reported in countries such as Denmark, Sweden, the Netherlands, Spain, Israel and China. These studies have shown non-secretors to be protected from infection by GII.4 noroviruses [42,49], albeit with some exceptions [43,50,51], which will be discussed separately.

Furthermore, several outbreak studies of the GII.3 genotype, which is one of the most common norovirus genotypes, have yielded somewhat conflicting results. Some studies indicated that non-secretors are less susceptible to infection, whereas other studies showed no association between secretor status and susceptibility [42,43,44,46].

Two studies involving the GI.3 genotype, one waterborne and one foodborne outbreak in the Netherlands and Sweden, respectively [12,39], showed that secretor status was not associated with susceptibility to symptomatic infection. However, individuals with blood group B exhibited a lower risk of contracting norovirus-associated gastroenteritis. 

Of particular interest is the recent emergence of the previously rare GII.17 genotype, especially in East Asia [52]. Although few in vivo studies with limited numbers have been conducted on genetic susceptibility, some of the available data indicate that the GII.17 genotype is also secretor-specific, similar to the GII.4 genotype [48,52].

To conclude, outbreak studies in various settings and different countries have, with some exceptions, shown a near complete protection of non-secretors against the globally dominant GII.4 genotype. These studies have also demonstrated that the GI.3 genotype infects both secretors and non-secretors, with blood type B leading to a partial protection, similar to that seen in challenge studies of the prototype Norwalk strain (GI.1 genotype).

## 7. Norovirus Observational Studies

In addition to outbreak and challenge studies, observational studies, including prospective active and/or passive surveillance studies and birth cohorts have been conducted regarding norovirus and host genetic susceptibility, mainly in pediatric populations. These types of studies capture a wider diversity of norovirus genotypes than challenge and outbreak studies, although it can be difficult to assess susceptibility patterns for individual genotypes due to the limited number of samples. Prospective studies in countries such as the USA, Ecuador, Burkina Faso, Vietnam, China and Nicaragua have further confirmed that non-secretors are almost completely resistant to the GII.4 genotype. These studies have also shown that different GII.4 variants exhibit the same secretor specificity (Table 3) [40,41,44,47,53]. A study in Vietnam further indicated that the common GII.3 genotype predominantly infected secretors, although the association was not as strong as for GII.4. Studies from Ecuador and the USA have further shown that the GII.6 genotype and a novel genotype (tentatively assigned GII.23) predominantly infect secretors [40,41]. 

Importantly, these studies have shown that a wide array of norovirus genotypes also infect non-secretors. Genotypes such as GI.3, GI.6, GII.1, GII.2 and GII.7 have all been found to infect non-secretors to a similar extent as secretors [40,41,47,55]. Indeed, a birth cohort study performed in Ecuador showed significantly higher rates of non-GII.4 norovirus infection in secretor-negative children [41], suggesting that the non-secretor genotype or phenotype also increases susceptibility to infection by certain norovirus genotypes, although this could also be due to the predominance of the GII.4 genotype in secretor populations.

To conclude, observational studies have demonstrated that the predominant GII.4 genotype, as well as some other less common genotypes, has clear secretor specificity. These studies have also shown a wide array of different norovirus genotypes that are able to infect non-secretors. To date, however, no genotype has been found to exclusively infect non-secretors.

## 8. Population Genetics, Infection Rates and Epidemiology

The genetic or phenotypic diversity of human HBGAs is highly dependent on ethnicity. The differential expression of HBGAs is associated with many infectious diseases [56], with the potential to be a strong evolutionary driver of the high level of genetic diversity between different population groups. Approximately 75–80% of Caucasian, Central Asian, and several African populations are secretors [57,58], while in Meso-American populations the prevalence of secretors can be as high as 95% [40,41,53]. Furthermore, the Lewis-negative phenotype is significantly more prevalent in several African countries (20–33%) than in European and some Asian populations (6–11%) [47,57,58]. Since different norovirus genotypes infect individuals of different HBGA phenotypes, this HBGA diversity is likely to influence norovirus epidemiology and infection rates at a population level. The globally predominant genotype GII.4 infects secretors almost exclusively. Moreover, secretors would be susceptible to both secretor and secretor-independent genotypes (Figure 2). These factors suggest that populations with a higher prevalence of secretors would have more infections, and perhaps a higher disease burden, especially since some studies have suggested that GII.4 cause more clinically severe disease [59,60,61].

Similarly, populations with a high diversity of HBGA phenotypes would likely be susceptible to a larger variety of different norovirus genotypes. HBGA diversity is higher in sub-Saharan African populations compared to most other regions [47,58] and, similarly, the norovirus genotype distribution in children is generally more diverse in Africa [58]. However, additional genetic studies from different regions, especially sub-Saharan Africa, are warranted.

Moreover, it is important to consider that secretor status is not binary. There are several missense mutations in the *FUT2* gene that result in reduced fucosyltransferase activity of the FUT2 enzyme (Table 1), and other factors such as microbiota that also affect glycosylation in the gut [62]. For example, in many East Asian populations, the complete non-secretor phenotype is rare or non-existent. Instead, the weak-secretor genotype, Se^385^Se^385^ (Table 1), which results in the reduced expression of secretor HBGAs, has approximately 15–20% prevalence [63] (Figure 2). Individuals with a weak-secretor phenotype could be susceptible to several different norovirus genotypes, both secretor-dependent and secretor-independent, because they express both secretor and non-secretor glycans (Figure 2). In accordance with this, children with a weak-secretor phenotype have been found to have partial, but not complete protection against GII.4 viruses [44], probably due to the lower expression of secretor glycans.

## 9. Norovirus GII.4 Infection in Non-Secretors

While most studies have shown that non-secretors are protected against GII.4 infection and disease, exceptions have been found [33,43,44,47,50,51]. First, it is important to consider that many studies describing GII.4 infections in non-secretors have been performed in East Asian populations, where the weak-secretor genotype is common (Table 1, Figure 2), thus possibly resulting in only partial resistance due to the low, but not absent, expression of the H antigen. For example, in a study of Chinese pediatric patients, the weak secretor genotype (385 (A>T)) provided significant but not complete protection against GII.3 and GII.4 viruses [44]. In the only challenge study performed with a GII.4 virus, the non-secretors were significantly protected although one non-secretor also became asymptomatically infected [33]. Furthermore, in a GII.4 virus outbreak in an elderly nursing home in Spain, one non-secretor was found to be infected [50]. Similarly, in a prospective study of children with diarrhea in Burkina Faso [47], one non-secretor child with diarrhea was found to be infected with the GII.4 genotype. In a study of norovirus outbreaks in long-term care facilities in the USA caused by GII.4 variants, GII.4 den Haag and GII.4 New Orleans, no association between secretor status and symptomatic infection was detected, and VLPs of these strains were also able to bind to Lewis-positive non-secretor saliva [11,51]. However, the number of infected non-secretors was low (Table 3).

A limitation of some of these studies was the use of a phenotypic determination of secretor status in saliva. Discrepancies between saliva and gut HBGA expression can occur, for example, due to infection with pathogens such as *Helicobacter pylori* [64]; thus, a genotypic determination of secretor status should be regarded as the most reliable determinant of secretor status. Furthermore, not all GII.4 variants have been investigated to the same extent, and some in vitro binding studies [11] and in vivo cases [51] suggest that there might be differences in secretor specificity between the GII.4 variants, although most studies show a similar secretor specificity among GII.4 variants in vivo (Table 3). It has been suggested that an extended HBGA binding spectrum observed for some of the more recent GII.4 variants may contribute to the epidemiological importance of these variants [11]. 

Thus, although non-secretors are generally protected against infection and disease caused by GII.4 genotypes, there are cases described when non-secretors become both asymptomatically and symptomatically infected. The reasons for this are as of yet unclear, but could include microbiota diversity, including HBGA-expressing bacteria [65], differences between GII.4 variants, general health status, weak-secretor phenotype, or other unidentified host factors.

## 10. Genetic Susceptibility and Asymptomatic Norovirus Infection

Norovirus is commonly found in asymptomatic healthy individuals, with a prevalence ranging from 5–15% [66,67]. The clinical outcome of norovirus infection is partly genotype-dependent, with studies demonstrating that the predominant GII.4 genotype causes more severe symptoms and is relatively less prevalent in asymptomatic controls compared to other genotypes [53,60,61,66]. The GII.4 genotype has also been reported to be shed in higher titers compared to other genotypes [59,68].

Studies of genetic susceptibility in asymptomatic individuals are rare since most studies adopt clinical symptoms as an outcome, thus excluding asymptomatic infections and hindering an assessment of the influence of HBGA genetics on both infection and clinical symptoms. In a challenge study using the Norwalk strain (GI.1 genotype) with the results stratified according to both symptomatic and asymptomatic infections, the authors concluded that non-secretors were protected against both symptomatic and asymptomatic infections [34]. In an active surveillance study conducted in the USA, both symptomatic and asymptomatic children infected with the GII.4 genotype were found to be secretors [40]. Moreover, a study of asymptomatic norovirus-positive children in Nicaragua [66] showed similar HBGA and secretor distribution in asymptomatic norovirus-positive children as compared to that identified in a previous study of symptomatic children conducted in the same country [53].

Thus, the studies performed to date suggest that both symptomatic and asymptomatic norovirus infection follow the same HBGA susceptibility patterns, with non-secretors resistant to both symptomatic and asymptomatic infections as a result of secretor-specific genotypes, such as the predominant GII.4.

## 11. The Lewis Antigens as Mediators of Susceptibility

The Lewis a or Lewis b antigen is commonly used as a surrogate marker for secretor status; however, few in vivo studies have been able to elucidate their role in susceptibility. This is mainly due to the rarity of the Lewis-negative genotype or phenotype in the populations in which most of the studies have been conducted (~3–8% in Caucasians and Asians). It is clear that the presence of an α1,2-linked fucose (secretors) is essential for susceptibility to a wide variety of norovirus genotypes. The presence of both α1,2-linked and α1,4-linked fucoses (Lewis b) has been suggested to have a synergistic effect on the susceptibility to some norovirus strains in vitro [38]. Furthermore, while specific binding to Lewis antigens by some norovirus genotypes has been demonstrated in vitro [11,38,69], in vivo studies supporting the Lewis antigens as direct mediators of susceptibility are rare. One prospective study of pediatric diarrhea in Burkina Faso, where the prevalence of the Lewis-negative phenotype is high (approximately one-third of the population) showed that G1 viruses infected only Lewis-positive and not Lewis-negative children, whereas GII viruses were found in both Lewis-negative and Lewis-positive children [47]. Another study from Nicaragua, where the Lewis-negative phenotype is also common, demonstrated that secretor and not Lewis status was the strongest mediator of susceptibility [53]. Moreover, a Swedish study showed that both Lewis-positive and Lewis-negative secretors had equal titers of antibodies against GII viruses [70]. Interestingly, Kubota et al. demonstrated a structural basis for the recognition of Lewis antigens by Genogroup 1 viruses [71], which is in accordance with the findings of the Burkina Faso study [47].

In vitro binding studies of the binding of recent GII.4 strains to the Lewis fucose in non-secretors have been reported [11,51,72], indicating that some GII.4 variants can infect non-secretors [51]. However, as mentioned previously in this review, reports of GII.4 in non-secretors are relatively rare, and almost all non-secretors in studies where this is reported are Lewis-positive (in total 90–95% of Caucasian populations are Lewis-positive). Furthermore, in a study in Burkina Faso, a GII.4 virus was detected in a Lewis-negative non-secretor, indicating that the presence of a Lewis antigen is not necessary to confer susceptibility to this GII.4 strain [47].

To conclude, the importance of Lewis antigens as mediators of susceptibility for many genotypes is unclear. There are some indications that they are important for susceptibility to GI viruses in vivo, but there is less evidence for this for GII viruses.

## 12. Norovirus GII.4 Evolution and Receptor Switching

Many studies have provided evidence that the predominant norovirus GII.4 genotype mainly infects secretors, and all GII.4 variants identified in the past show similar secretor affinity (Table 3). However, in vitro binding studies have shown that the major GII.4 variants bind to the major secretor antigens with varied binding affinities, and also to glycans present in non-secretors [11,24,51,73]. Furthermore, serum blocking titers vary with VLPs of different GII.4 variants [24,74]. The protruding parts of the norovirus capsid, which contains both antigenic and carbohydrate-binding ligands, are subject to heavy immune selection. It has been suggested that norovirus GII.4 persists not only through altered antigenicity resulting from genetic drift, but also by altering their HBGA carbohydrate-binding targets over time. By changing receptor or attachment factors, it can be speculated that norovirus escapes protective herd immunity, allowing the infection of an immunologically naïve population [24,75]. 

Only a few in vivo susceptibility studies support the hypothesis that different GII.4 strains or variants infect individuals with different HBGAs. Halperin et al. studied two GII.4 outbreaks caused by two different strains in military units in Israel. In the first outbreak, no association was found between different blood groups, whereas, in the second outbreak, individuals with blood groups A or AB had a decreased risk of infection [76]. Moreover, a study in Burkina Faso showed that GII.4 genotypes preferentially infected secretors with blood group B [47], whereas another study in China showed that GII.4 genotypes preferentially infected secretors with blood group A [42]. However, in several other studies, no association between secretor antigens (ABO or Lewis) and susceptibility to different GII.4 variants was identified. In fact, the ability of GII.4 genotypes to infect individuals with different secretor antigens, and varying levels of expression, broadens the pool of susceptible individuals and may be a factor accounting for their predominance in the human population.

Thus, a few in vivo studies support the hypothesis that different GII.4 variants have different secretor antigen affinity, but additional in vivo studies are required to clarify the ability of different secretor antigens, such as ABO, to affect susceptibility to GII.4 genotypes in a strain- or variant-dependent manner.

## 13. Genetics vs. Immunity—Prolonged Norovirus Infection in Immunocompromised Hosts

In healthy individuals, symptoms generally last for 2–3 days and the infection and norovirus shedding are cleared within 21 days [77]. However, prolonged or even chronic norovirus infection can occur in individuals with primary or secondary immunodeficiency [78]. Chronic norovirus infection has been reported in several studies, most of which were associated with solid organ transplants [79,80,81], but also allogenic stem cell transplants [82,83], and patients with common variable immunodeficiency (CVID) [78]. The symptoms vary, from prolonged diarrhea to asymptomatic with recurrent diarrhea, with or without vomiting [79,84]. 

A better understanding of the role of genetics in controlling HBGA expression, in the context of the immunology of infection susceptibility, may be gained through studies of norovirus infection in immunocompromised individuals in relation to genetic factors.

As for acute norovirus infection, most studies describing chronic norovirus infection have been due to GII.4 genotypes, although other genotypes have also been reported [14,81,85]. However, few studies have focused on the association of HBGA phenotypes or genotypes with chronic norovirus infection, and those that have were mainly conducted in a single individual. Individual studies of chronic GII.4 infection in a kidney transplant recipient [80] and a GII.3 infection in a heart transplant recipient [14] showed that the patients were secretors.

In vitro binding studies have been performed using VLPs developed from chronically infected GII.4 strains. These studies showed that the VLPs have similar secretor-specific binding profiles and that the HBGA binding epitopes are generally well conserved and maintained over time in the chronically infected [84,86], although some changes have been observed [87].

In a prospective study, six individuals infected with norovirus of the GI.3 genotype following allogenic stem cell transplantation were all secretors [82]. Previous studies have indicated that the GI.3 genotype is secretor-independent [12,39] (Table 2), although this could not be corroborated due to the low number of samples. Another outbreak of the GII.4 genotype occurred in 11 immunosuppressed individuals (due to chemotherapy or stem cell transplantation) who were all secretors, which was also consistent with the secretor profile of the GII.4 genotype. However, the number of patients in this study was also too few for reliable conclusions to be drawn [49]. While all these studies investigated the secretor genotype or phenotype of the stem cell recipient, no information is yet available regarding the influence of the secretor status of the stem cell donors on the outcome of the infection in the recipient. Such studies on this important issue should be given priority.

Thus, there is a need for more in vivo studies to clarify whether the same genetic mechanisms determining norovirus susceptibility in immunocompetent individuals are also valid in immunocompromised patients. However, the wide range of different immunodeficiencies, which could impact the results and comparisons between different studies, remains a challenge.

## 14. Human Intestinal Enteroids: A Novel Model to Study Genetic Susceptibility to Norovirus

For decades, there has been a lack of a robust in vitro system that supports norovirus replication and propagation. Recently, human intestinal enteroids (HIEs), derived from human small intestinal tissue, were used successfully to support human norovirus replication [88,89]. Moreover, norovirus has also been propagated successfully in human-induced pluripotent stem cell-derived intestinal epithelial cells [90]. Ettayebi et al. [88] generated HIEs from jejunal samples obtained from individuals genotyped as secretors and non-secretors. Two GII.4 genotype noroviruses (variants Den Haag 2006b and Sydney 2012) replicated successfully in all secretor-positive HIEs, but not in HIEs from non-secretors. The GII.3 genotype also infected all secretor-positive HIEs and, to a lesser extent, non-secretor HIEs. Thus, these results are in accordance with the in vivo data (Table 2 and Table 3) and show that the HIEs are a relevant in vitro system for studies of genetic susceptibility to norovirus. Future studies with different norovirus genotypes or variants using HIEs of different secretor, Lewis and ABO profiles are required to answer many outstanding questions regarding genetic susceptibility and genotype-specific infection.

## 15. Impact of Secretor Status on Norovirus Vaccine Design

At least six VLP-based norovirus vaccine candidates are, or have been, in development [10], with two in clinical trials. Most candidates are based on the prototype GI.1 or GII.4 genotypes, both of which show strong secretor specificity. As most VLP-based vaccines are designed for parenteral administration [10], thereby bypassing the mucosa, genetic differences related to HBGA expression in the small intestine are unlikely to have a large impact on vaccine immunogenicity. For example, non-secretors appear to have a similar total immunoglobulin response to a single dose of a parenteral vaccine containing VLPs from both GI.1 and GII.4 norovirus as compared to secretors [91].

The recent success in propagating norovirus in vitro [88] bodes well for the development of oral live norovirus vaccines. If such a candidate were developed, it would be important to characterize HBGA expression in the recipient in relation to the genotype(s) of the vaccine strain. The vaccine-take rate and subsequent protection is likely to be influenced if the vaccine recipient is resistant to the live vaccine strain, as has been observed for the live oral rotavirus vaccines [92,93]. An optimal live vaccine candidate should contain viruses of the predominant GII.4 genotype in addition to secretor-independent genotypes such as GI.3, thus covering both genogroups and both secretor profiles.

## 16. Relationships between Secretor Status, Microbiota and Susceptibility to Norovirus Infection

Recent advances point to the influence of the gut microbiome on norovirus infection. While fucosylation by the FUT2 enzyme is known to be important for norovirus susceptibility, commensal bacteria in the intestine may also influence infection by modulating glycosylation of the intestinal mucosa [94,95]. Thus, these changes might directly affect norovirus susceptibility by altering norovirus adherence to specific glycans.

In vitro experiments with norovirus VLPs (GI.1 and GII.6 genotypes) have demonstrated the ability of *Enterobacter* sp. to bind viral particles due to the presence of HBGA-like glycans of the A, B and H types [65], and that the norovirus P particles of the GI.1 and GII.4 genotypes have the capacity to interact with enteric commensals [96]. Rodrigues-Diaz and co-workers [97] found that individuals with a greater abundance of *Rumininococcacaea* and *Faecalibacterium* bacteria might have lower susceptibility to norovirus infection by measuring salivary IgA titers in 35 adult individuals. However, it still remains unclear if the microbiota affects susceptibility of norovirus infection, particularly to the globally dominating GII.4 virus. The successful infection of human GII.4 virus in stem cell-derived intestinal enteroids has recently shown that commensal bacteria is not required for infection [88] and that enteroids from non-secretors are resistant to GII.4 infections, altogether further supporting that secretor status is a strong restriction factor for GII.4 virus infection.

As secretor status is currently the strongest correlate of susceptibility by the predominant GII.4 norovirus, it is important to understand whether secretor status also affects the microbiota composition which could then indirectly affect norovirus susceptibility. Although some, in general smaller, studies have found secretor status to influence microbiota [98], recent larger studies have found no such association. A study powered to detect the association of *FUT2* with microbial composition including 1190 healthy individuals found no association between the FUT2 genotype and inferred phenotype and human fecal microbial composition [99]. These results are in agreement with another recent study using a large cohort of 1503 twins [100], as well as different genome-wide association studies that found no or weak association of *FUT2* and fecal microbiome composition [101,102]. 

In conclusion, a direct link between microbiota and susceptibility to norovirus infection remains elusive. The interactions between virus, host, glycans and microbiota are complex and need further investigation. Outbreak or challenge studies with a single norovirus genotype, where microbiota composition, adjusted for HBGA genotypes, is investigated in terms of susceptibility, might shed more light on the topic.

## 17. Conclusions

While the strong Mendelian resistance to GII.4 genotype noroviruses among individuals with a nonsense mutation in the *FUT2* gene has contributed to our understanding of the restriction pattern of norovirus infection, more studies are needed to understand the host restriction pattern for several less common norovirus genotypes. Despite using Lewis antigens as biomarkers of secretor status, our current knowledge of their role in this process is limited, and based mainly on in vitro binding studies. Furthermore, few studies have examined norovirus infection in immunodeficient individuals in relation to HBGAs and further studies are also needed to clarify the role of microbiota in relation to *FUT2* polymorphisms. The HBGA phenotype varies greatly between different populations, which is likely to affect both norovirus epidemiology and disease burden at a population level; therefore, additional genetic studies in different populations and coupled with epidemiology and disease data are warranted to address this issue. The human enteroid model mimics the human intestinal epithelium; therefore, a library of enteroids generated from individuals with different *FUT2*, *FUT3* and *ABO* polymorphisms may provide important information on how secretor status and Lewis glycans, as well as other HBGAs, affect strain-specific human norovirus infection. 

## Figures and Tables

**Figure 1 viruses-11-00226-f001:**
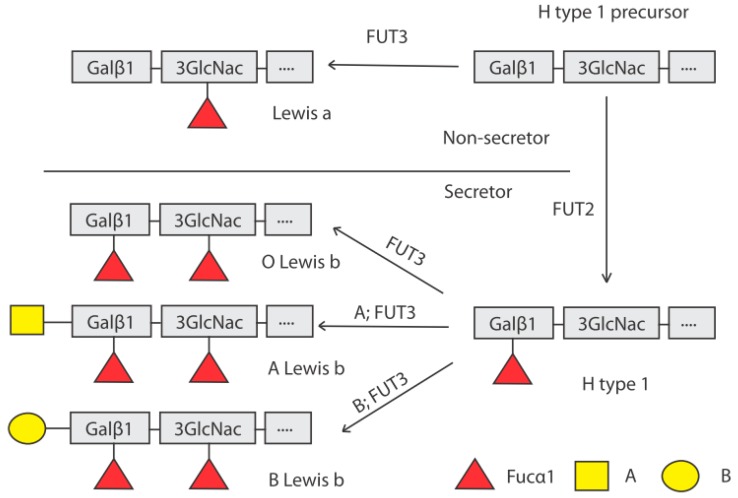
Biosynthesis pathway of histo-blood group antigens (HBGAs) by stepwise addition of monosaccharides to precursor structures. The *FUT2* (secretor) gene encodes an α1,2-fucosyltransferase which adds a fucose residue in α1,2 linkage to the terminal galactose of the H type 1 precursor. Non-secretors, having a non-functional FUT2 enzyme, cannot synthesize the H type 1 antigen from its precursor and lack α1,2-linked fucose HBGAs in the intestinal mucosa and other secretions. The synthesis of the A and B antigens by the corresponding enzymes requires the presence of the H type 1 antigen, adding an *N*-acetylgalactosamine (A) or a galactose (B) in a α1,4 linkage on the galactose residue of the H type 1 antigen. The Lewis antigens are synthesized with the FUT3 enzyme which adds a α1,4-linked fucose residue on the N-acetylglucosamine of the H antigen or the H antigen precursor, generating Lewis a or Lewis b antigens respectively. Abbreviations: Gal: galactose; GlcNAc: *N*-acetylglucosamine; Fuc: fucose.

**Figure 2 viruses-11-00226-f002:**
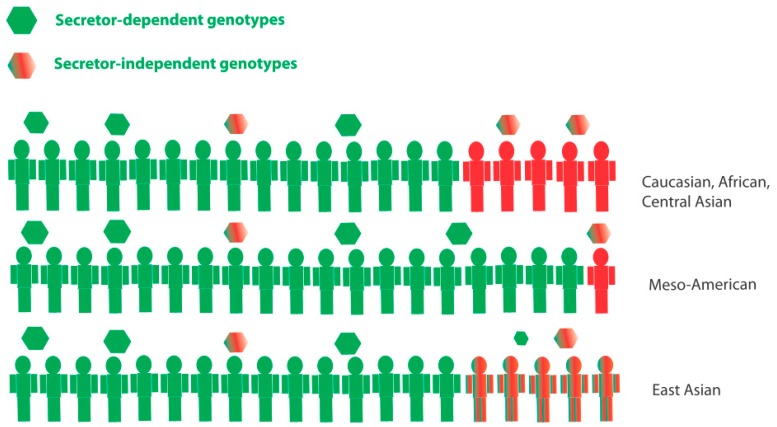
A simplistic overview of secretor status in different population groups. Secretors (green) are susceptible to secretor-dependent norovirus genotypes, including the globally predominant GII.4 genotype. Secretors are also further susceptible to secretor-independent genotypes. Non-secretors (red) are only susceptible to secretor-independent genotypes. As of yet, no genotypes have been observed to exclusively infect non-secretors. Weak-secretors (mix red or green) are common in East Asian populations, having a reduced expression of secretor antigens, and would thus be partially susceptible to secretor-specific genotypes. The prevalence of secretors is particularly high in Meso-American populations, which may lead to a higher disease burden in the population due to (1) higher rates of GII.4 infection and (2) a larger portion of the population being susceptible to both secretor and secretor-independent genotypes.

**Table 1 viruses-11-00226-t001:** *FUT2* polymorphisms with known effect on secretor status in different populations. Adapted from [31]. The two most common and investigated in genetic susceptibility studies are highlighted in bold.

Allele	Mutation Type	Phenotype	Population
Se^302^	Missense	Non-secretor ^a^	Thai, Bangladeshi
**Se^385^**	Missense	Weak-secretor	East Asian
**Se^428^**	Nonsense	Non-secretor	Caucasian, African, Meso-American, Central Asian
Se^571^	Nonsense	Non-secretor	Filipino and Samoan
Se^628^	Nonsense	Non-secretor	Japanese and Chinese
Se^658^	Nonsense	Non-secretor	Chinese
Se^357,480,778^	Frameshift	Non-secretor	African
Se^849^	Nonsense	Non-secretor	Chinese and Filipino
Se^del1^	Deletion	Non-secretor	Indian and Bangladeshi
Se^del2^	Deletion	Non-secretor	Samoan
Se^fus^	Fusion	Weak-secretor	Japanese

^a^ No α1,2-fucosyltransferase activity found in transfected COS7 cells [32].

**Table 2 viruses-11-00226-t002:** Distribution of ABO and secretor status in studies with GI and non-GII.4 genotype infections (symptomatic and/or asymptomatic). Only included are studies and/or genotypes with at least five representative samples.

Genotype	Country		O	A	B	AB	Secretor	Non-Secretor	Type of Study	Reference
**GI**										
GI.1	USA	21/28 (75%)		10/19 (53%)	3/7 (43%)	0/1 (0%)	34/55 (62%)	0/22 (0%)	Challenge	[28]
GI.1 ^a^	USA		25/26 (96%)	14/18 (78%)	3/5 (60%)	0/2 (0%)	42/43 (98%)	0/8 (0%)	Challenge	[34,35,37]
GI.3	Sweden		10/23 (43%)	14/32 (44%)	2/12 (17%)	0/1 (0%)	26/68 (38%)	7/15 (47%)	Outbreak	[12]
GI.3	Netherlands		11/11 (100%)	9/9 (100%)	0/2 (0%)	0/0 (0%)	20/22 (91%)	4/7 (57%)	Outbreak	[39]
**GII-nonGII.4**										
GII.1	USA		N.A	N.A	N.A	N.A	+ ^c^	+	Active surveillance	[40]
GII.1	Ecuador		N.A	N.A	N.A	N.A	+	+	Birth cohort	[41]
GII.2 ^a^	USA		4/8 (50%)	2/4 (50%)	1/1 (100%)	2/2 (100%)	8/12 (67%)	1/3 (33%)	Challenge	[36]
GII.3 ^b^	China		3/6 (50%)	5/8 (63%)	1/2 (50%)	0/0 (0%)	9/14 (64%)	0/2 (0%)	Outbreak	[42]
GII.3 ^b^	China		8/22 (36%)	4/6 (67%)	4/8 (50%)	3/3 (100%)	18/38 (47%)	1/1 (100%)	Outbreak	[43]
GII.3 ^b^	China		N.A	N.A	N.A	N.A	15/102 (15%)	1/22 (5%)	Active surveillance	[44]
GII.3	Tunisia		+	+	+	+	23/76 (30%)	5/22 (23%)	Passive surveillance	[45]
GII.3 ^d^	Vietnam		N.A	N.A	N.A	N.A	23/229 (10%)	5/31 (16%)	Surveillance	[46]
GII.6	USA		N.A	N.A	N.A	N.A	+	+/−	Active surveillance	[40]
GII.7	USA		N.A	N.A	N.A	N.A	+	+	Active surveillance	[40]
GII.10	Burkina Faso		4/71 (6%)	1/34 (3%)	0/55 (0%)	0/4 (0%)	5/164 (3%)	0/44 (0%)	Passive surveillance	[47]
GII.17	China		23/69 (33%)	23/46 (50%)	18/47 (38%)	3/7 (43%)	67/169 (40%)	2/23 (9%)	Outbreak	[48]
GII.23	Ecuador		N.A	N.A	N.A	N.A	+	−	Birth cohort	[41]

^a^ Blood groups determined for both secretors and non-secretors; ^b^ Study in populations where the weak secretor genotype is common; ^c^ When real numbers are not publicly available; indicated by susceptible (+) resistant (−) reduced susceptibility (+/−); ^d^ Secretor and/or partial secretor.

**Table 3 viruses-11-00226-t003:** Distribution of ABO and secretor status in studies with GII.4 genotype infections (symptomatic and asymptomatic). Only included are studies and/or genotypes with at least five representative samples.

GII.4 Variant	Country	O	A	B	AB	Secretor	Non-Secretor	Type of Study	Reference
New Orleans 2009	USA	N.A	N.A	N.A	N.A	+ ^a^	−	Active surveillance	[40]
Den Haag 2006b	USA	N.A	N.A	N.A	N.A	+	−	Active surveillance	[40]
Sydney 2012	USA	N.A	N.A	N.A	N.A	+	−	Active surveillance	[40]
Den Haag 2006b	USA	+	+	+	+	15/25 (60%)	1/2 (50%)	Outbreaks	[51]
New Orleans 2009	USA	+	+	+	+	39/49 (80%)	2/3 (66%)	Outbreaks	[51]
Sydney 2012	USA	+	+	+	+	5/9 (55%)	0/2 (0%)	Outbreaks	[51]
New Orleans 2009 ^b^	Ecuador	N.A	N.A	N.A	N.A	+	−	Birth cohort	[41]
Den Haag 2006b ^b^	Ecuador	N.A	N.A	N.A	N.A	+	−	Birth cohort	[41]
Farmington Hills 2002	USA	NA	NA	NA	NA	16/23 (70%)	1/17 (6%)	Challenge	[33]
Den Haag 2006b	China	N.A	N.A	N.A	N.A	17/102 (17%)	1/22 (5%)	Active surveillance	[44]
Hunter 2004	Spain	N.A	N.A	N.A	N.A	33/43 (77%)	1/17 (6%)	Outbreak	[50]
N.A	Denmark	N.A	N.A	N.A	N.A	29/52 (56%)	0/9 (0%)	Outbreak	[30]
N.A	Sweden	N.A	N.A	N.A	N.A	53/97 (55%)	0/18 (0%)	Outbreaks	[49]
N.A	China	7/50 (14%)	25/52 (48%)	9/28 (32%)	3/5 (60%)	41/115 (36%)	0/15 (0%)	Outbreak	[42]
Den Haag 2006b ^c^	China	24/35 (69%)	20/28 (71%)	7/9 (78%)	2/2 (100%)	49/69 (71%)	4/5 (80%)	Outbreak	[43]
Mixed ^d,e^	Vietnam	N.A	N.A	N.A	N.A	22/229 (10%)	0/31 (0%)	Surveillance	[46]
Mixed	Burkina Faso	1/71 (1%)	0/34 (0%)	7/55 (13%)	1/4 (25%)	9/164 (5%)	1/44 (2%)	Passive surveillance	[47]
Mixed	Nicaragua	+	+	+	− ^f^	+	− ^f^	Passive surveillance	[53]
Unknown ^c^	Israel	16/30 (53%)	17/34 (50%)	7/16 (44%)	6/11 (55%)	N.A	N.A	Outbreak	[54]
Unknown ^c^	Israel	36/66 (55%)	28/77 (36%)	21/45 (47%)	7/25 (28%)	N.A	N.A	Outbreak	[54]

^a^ When real numbers are not publicly available; indicated by susceptible (+) resistant (−) reduced susceptibility (+/−); ^b^ The study also included Sydney (*n* = 4) and Osaka (*n* = 1) GII.4 variants; ^c^ Blood groups determined in both secretor and non-secretors; ^d^ Secretor and/or partial secretor included in secretor group; ^e^ Including Den Haag 2006b, New Orleans 2009 and Sydney 2012 variants; ^f^ Prevalence of AB and non-secretor is low in Nicaragua.

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
