# Peer review of "Genetic Susceptibility to Human Norovirus Infection: An Update"

_viruses, 2019, doi:10.3390/v11030226_

Round 1

Reviewer 1 Report

Nordgren and Svensson have compiled a comprehensive review on human norovirus genetic susceptibility and HBGA expression, including data from challenge studies and outbreaks, symptomatic and asymptomatic infections.  The manuscript is well written.  Complex data sets are presented in clear, concise tables.   My minor comments are listed below.

        Line 79, in response to the conclusion from the early challenge studies, that there is no long-term immunity to infection, the data from this study has been reconsidered given the dose used compared to the virus ID50. It has been suggested that the virus challenge load may have overwhelmed any immunity.  Long-term immunity is supported by modeling, human monoclonal antibodies, antigenic seniority and viral evolution.  Please modify.

2.       Section 4.  It may be informative to readers to add a sentence about the connection between HBGA expression in mucosa and blood type.

3.       Section 5. Lines 141, How is challenge with less-common GI.1 or GII.2 strains is not clinically relevant but challenge with an extinct virus (GII.4 2002) relevant?   Please explain or modify.

4.       Table 2: On the PDF, some of the text is not aligned and I am unable to find superscript c within the table.

5.       Table 3: PDF has some entries misaligned.  Superscript c is out of alphabetical order.

6.       Section 12: My understanding of the literature on GII.4 strain variation and HBGA affinity is that sequence variation in the residues around the carbohydrate binding pocket stabilizes interaction with different HBGAs making binding more or less likely, not yes verses no binding or preferential infection of one HBGA type over another (excluding secretor status).  This would likely broaden the susceptibility pool not limit the pool to a specific HBGA profile.  Section 12 states there is little support from outbreak data that different GII.4 strains selectively infect a defined HBGA profile and this clear from the data.  These data do support the hypothesis that breadth of GII.4 binding leads to infection of more individuals with varying levels and profiles of HBGA expression.  Please clarify.

7.       References: DOI are not included.

Author Response

 Line 79, in response to the conclusion from the early challenge studies, that there is no long-term immunity to infection, the data from this study has been reconsidered given the dose used compared to the virus ID50. It has been suggested that the virus challenge load may have overwhelmed any immunity.  Long-term immunity is supported by modeling, human monoclonal antibodies, antigenic seniority and viral evolution.  Please modify.

Answer: This has been modified as suggested, with an additional reference added

2.       Section 4.  It may be informative to readers to add a sentence about the connection between HBGA expression in mucosa and blood type.

Answer: A sentence regarding blood type and HBGA expression has been added as suggested

3.       Section 5. Lines 141, How is challenge with less-common GI.1 or GII.2 strains is not clinically relevant but challenge with an extinct virus (GII.4 2002) relevant?   Please explain or modify.

Answer: More discussion regarding this has been added as suggested

4.       Table 2: On the PDF, some of the text is not aligned and I am unable to find superscript c within the table.

Answer: The texts on the table is now aligned, and superscript c is added.

5.       Table 3: PDF has some entries misaligned.  Superscript c is out of alphabetical order.

Answer: The text has now been aligned, and the order of the superscripts checked.

6.       Section 12: My understanding of the literature on GII.4 strain variation and HBGA affinity is that sequence variation in the residues around the carbohydrate binding pocket stabilizes interaction with different HBGAs making binding more or less likely, not yes verses no binding or preferential infection of one HBGA type over another (excluding secretor status).  This would likely broaden the susceptibility pool not limit the pool to a specific HBGA profile.  Section 12 states there is little support from outbreak data that different GII.4 strains selectively infect a defined HBGA profile and this clear from the data.  These data do support the hypothesis that breadth of GII.4 binding leads to infection of more individuals with varying levels and profiles of HBGA expression.  Please clarify.

Answer: This has now been clarified as suggested.

7.       References: DOI are not included.

 Answer: For editorial office: Unsure if this is needed at this point? We are using endnote style as specified for the journal

Reviewer 2 Report

This is an extensive and timely review on an important topic to the field of genetics of susceptibility to infectious diseases. It is very well written and organized, as well as comprehensive with no omission of previously reported data.

Minor points:

- In section 9, norovirus infection in non-secretors, the authors state that differences between GII.4 variant may account for the reported infection of some non-secretors. Indeed this has been demonstrated as in reference 11, where the relative affinity of variants evolved. Variants with the highest relative affinity showing an extended spectrum of recognized glycans and consequently of hosts. This would be worth mentioning.

- Tables 2 and 3 require some formating

- References should be checked, i.e. reference 47 is incomplete

Author Response

This is an extensive and timely review on an important topic to the field of genetics of susceptibility to infectious diseases. It is very well written and organized, as well as comprehensive with no omission of previously reported data.

Minor points:

- In section 9, norovirus infection in non-secretors, the authors state that differences between GII.4 variant may account for the reported infection of some non-secretors. Indeed this has been demonstrated as in reference 11, where the relative affinity of variants evolved. Variants with the highest relative affinity showing an extended spectrum of recognized glycans and consequently of hosts. This would be worth mentioning.

Answer: This has now been mentioned in the text as suggested

- Tables 2 and 3 require some formatting

Answer: The text in the tables have now been better aligned, as also suggested by reviewer 1

- References should be checked, i.e. reference 47 is incomplete

Answer: All references have now been checked and modified if incomplete